# Reinforced Continual Learning

**Ju Xu**
Center for Data Science, Peking University
Beijing, China
xuju@pku.edu.cn

**Zhanxing Zhu** *
Center for Data Science, Peking University &
Beijing Institute of Big Data Research (BIBDR)
Beijing, China
zhanxing.zhu@pku.edu.cn

## Abstract

Most artificial intelligence models are limited in their ability to solve new tasks faster, without forgetting previously acquired knowledge. The recently emerging paradigm of continual learning aims to solve this issue, in which the model learns various tasks in a sequential fashion. In this work, a novel approach for continual learning is proposed, which searches for the best neural architecture for each coming task via sophisticatedly designed reinforcement learning strategies. We name it as Reinforced Continual Learning. Our method not only has good performance on preventing catastrophic forgetting but also fits new tasks well. The experiments on sequential classification tasks for variants of MNIST and CIFAR-100 datasets demonstrate that the proposed approach outperforms existing continual learning alternatives for deep networks.

## 1 Introduction

Continual learning, or lifelong learning [15], the ability to learn consecutive tasks without forgetting how to perform previously trained tasks, is an important topic for developing artificial intelligence. The primary goal of continual learning is to overcome the forgetting of learned tasks and to leverage the earlier knowledge for obtaining better performance or faster convergence/training speed on the newly coming tasks.

In the deep learning community, two groups of strategies have been developed to alleviate the problem of forgetting the previously trained tasks, distinguished by whether the network architecture changes during learning.

The first category of approaches maintain a fixed network architecture with large capacity. When training the network for consecutive tasks, some regularization term is enforced to prevent the model parameters from deviating too much from the previous learned parameters according to their significance to old tasks [4, 19]. In [6], the authors proposed to incrementally match the moment of the posterior distribution of the neural network which is trained on the first and the second task, respectively. Alternatively, an episodic memory [7] is budgeted to store the subsets of previous datasets, and then trained together with the new task. FearNet [3] mitigates catastrophic forgetting by consolidating recent memories into long-term storage using pseudorehearsal [10] which employs a generative autoencoder to generate previously learned examples that are replayed alongside novel information during consolidation. Fernando et al. [2] proposed PathNet, in which a neural network has ten or twenty modules in each layer, and three or four modules are picked for one task in each layer by an evolutionary approach. However, these methods typically require unnecessarily large-capacity networks, particularly when the number of tasks is large, since the network architecture is never dynamically adjusted during training.

---

The other group of methods for overcoming catastrophic forgetting dynamically expand the network to accommodate the new coming task while keeping the parameters of previous architecture unchanged. Progressive networks [11] expand the architectures with a fixed size of nodes or layers, leading to an extremely large network structure particularly faced with a large number of sequential tasks. The resultant complex architecture might be expensive to store and even unnecessary due to its high redundancy. Dynamically Expandable Network (DEN, [17] alleviated this issue slightly by introducing group sparsity regularization when adding new parameters to the original network; unfortunately, there involves many hyperparameters in DEN, including various regularization and thresholding ones, which need to be tuned carefully due to the high sensitivity to the model performance.

In this work, in order to better facilitate knowledge transfer and avoid catastrophic forgetting, we propose a novel framework to adaptively expand the network. Faced with a new task, deciding optimal number of nodes/filters to add for each layer is posed as a combinatorial optimization problem. We provide a sophisticatedly designed reinforcement learning method to solve this problem. Thus, we name it as Reinforced Continual Learning (RCL). In RCL, a controller implemented as a recurrent neural network is adopted to determine the best architectural hyper-parameters of neural networks for each task. We train the controller by an actor-critic strategy guided by a reward signal deriving from both validation accuracy and network complexity. This can maintain the prediction accuracy on older tasks as much as possible while reducing the overall model complexity. To the best of our knowledge, the proposal is the first attempt that employs the reinforcement learning for solving the continual learning problems.

RCL not only differs from adding a fixed number of units to the old network for solving a new task [11], which might be suboptimal and computationally expensive, but also distinguishes from [17] as well that performs group sparsity regularization on the added parameters. We validate the effectiveness of RCL on various sequential tasks. And the results show that RCL can obtain better performance than existing methods even with adding much less units.

The rest of this paper is organized as follows. In Section 2, we introduce the preliminary knowledge on reinforcement learning. We propose the new method RCL in Section 3, a model to learn a sequence of tasks dynamically based on reinforcement learning. In Section 4, we implement various experiments to demonstrate the superiority of RCL over other state-of-the-art methods. Finally, we conclude our paper in Section 5 and provide some directions for future research.

## 2    Preliminaries of Reinforcement learning

Reinforcement learning [13] deals with learning a policy for an agent interacting in an unknown environment. It has been applied successfully to various problems, such as games [8, 12], natural language processing [18], neural architecture/optimizer search [20, 1] and so on. At each step, an agent observes the current state $s_t$ of the environment, decides of an action $a_t$ according to a policy $\pi(a_t|s_t)$, and observes a reward signal $r_{t+1}$. The goal of the agent is to find a policy that maximizes the expected sum of discounted rewards $R_t$, $R_t = \sum_{t'=t+1}^{\infty} \gamma^{t'-t-1} r_{t'}$, where $\gamma \in (0, 1]$ is a discount factor that determines the importance of future rewards. The value function of a policy $\pi$ is defined as the expected return $V_\pi(s) = E_\pi[\sum_{t=0}^{\infty} \gamma^t r_{t+1}|s_0 = s]$ and its action-value function as $Q_\pi(s, a) = E_\pi[\sum_{t=0}^{\infty} \gamma^t r_{t+1}|s_0 = s, a_0 = a]$.

Policy gradient methods address the problem of finding a good policy by performing stochastic gradient descent to optimize a performance objective over a given family of parametrized stochastic policies $\pi_\theta(a|s)$ parameterized by $\theta$. The policy gradient theorem [14] provides expressions for the gradient of the average reward and discounted reward objectives with respect to $\theta$. In the discounted setting, the objective is defined with respect to a designated start state (or distribution) $s_0$: $\rho(\theta, s_0) = E_{\pi_\theta}[\sum_{t=0}^{\infty} \gamma^t r_{t+1}|s_0]$. The policy gradient theorem shows that:

$$\frac{\partial \rho(\theta, s_0)}{\partial \theta} = \sum_s \mu_{\pi_\theta}(s|s_0) \sum_a \frac{\partial \pi_{\pi_\theta}(a|s)}{\partial \theta} Q_{\pi_\theta}(s, a). \tag{1}$$

where $\mu_{\pi_\theta}(s|s_0) = \sum_{t=0}^{\infty} \gamma^t P(s_t = s|s_0)$.

# 3 Our Proposal: Reinforced Continual Learning

In this section, we elaborate on the new framework for continual learning, Reinforced Continual Learning(RCL). RCL consists of three networks, *controller*, *value network*, and *task network*. The controller is implemented as a Long Short-Term Memory network (LSTM) for generating policies and determining how many filters or nodes will be added for each task. We design the value network as a fully-connected network, which approximates the value of the state. The task network can be any network of interest for solving a particular task, such as image classification or object detection. In this paper, we use a convolutional network (CNN) as the task network to demonstrate how RCL adaptively expands this CNN to prevent forgetting, though our method can not only adapt to convolutional networks, but also to fully-connected networks.

## 3.1 The Controller

Figure 1(a) visually shows how RCL expands the network when a new task arrives. After the learning process of task $t - 1$ finishes and task $t$ arrives, we use a controller to decide how many filters or nodes should be added to each layer. In order to prevent semantic drift, we withhold modification of network weights for previous tasks and only train the newly added filters. After we have trained the model for task $t$, we timestamp each newly added filter by the shape of every layer. During the inference time, each task only employs the parameters introduced in stage $t$, and does not consider the new filters added in the later tasks to prevent the caused semantic drift.

Suppose the task network has $m$ layers, when faced with a newly coming task, for each layer $i$, we specify the the number of filters to add in the range between 0 and $n_i - 1$. A straightforward idea to obtain the optimal configuration of added filters for $m$ layers is to traverse all the combinatorial combinations of actions. However, for an $m$-layer network, the time complexity of collecting the best action combination is $\mathcal{O}(\prod_1^m n_i)$, which is NP-hard and unacceptable for very deep architectures such as VGG and ResNet.

To deal with this issue, we treat a series of actions as a fixed-length string. It is possible to use a controller to generate such a string, representing how many filters should be added in each layer. Since there is a recurrent relationship between consecutive layers, the controller can be naturally designed as a LSTM network. At the first step, the controller network receives an empty embedding as input (i.e. the state $s$) for the current task, which will be fixed during the training. For each task $t$, we equip the network with softmax output, $\mathbf{p}_{t,i} \in \mathbb{R}^{n_i}$ representing the probabilities of sampling each action for layer $i$, i.e. the number of filters to be added. We design the LSTM in an autoregressive manner, as Figure 1(b) shows, the probability $p_{t,i}$ in the previous step is fed as input into the next step. This process is circulated until we obtain the actions and probabilities for all the $m$ layers. And the policy probability of the sequence of actions $a_{1:m}$ follows the product rule,

$$\pi(a_{1:m}|s;\theta_c) = \prod_{i=1}^{m} p_{t,i,a_i}, \tag{2}$$

where $\theta_c$ denotes the parameters of the controller network.

## 3.2 The Task Network

We deal with $T$ tasks arriving in a sequential manner with training dataset $\mathcal{D}_t = \{x_i, y_i\}_{i=1}^{N_t}$, validation dataset $\mathcal{V}_t = \{x_i, y_i\}_{i=1}^{M_t}$, test dataset $\mathcal{T}_t = \{x_i, y_i\}_{i=1}^{K_t}$ at time $t$. For the first task, we train a basic task network that performs well enough via solving a standard supervised learning problem,

$$\min_{W_1} L_1(W_1; \mathcal{D}_1). \tag{3}$$

We define the well-trained parameters as $W_t^a$ for task $t$. When the $t$-th task arrives, we already know the best parameters $W_{t-1}^a$ for task $t - 1$. Now we use the controller to decide how many filters should be added to each layer, and then we obtain an expanded child network, whose parameters to be learned are denoted as $W_t$ (including $W_{t-1}^a$). The training procedure for the new task is as follows, keeping $W_{t-1}^a$ fixed and only back-propagating the newly added parameters of $W_t \backslash W_{t-1}^a$. Thus, the optimization formula for the new task is,

$$\min_{W_t \backslash W_{t-1}^a} L_t(W_t; \mathcal{D}_t). \tag{4}$$

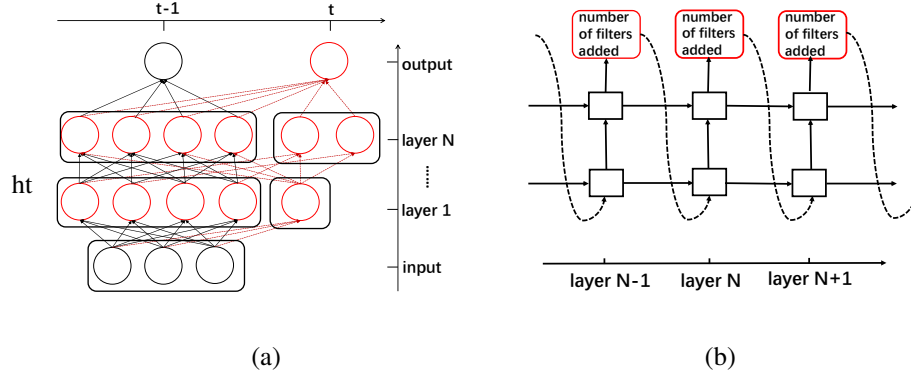

(a)                                    (b)

Figure 1: (a) RCL adaptively expands each layer of the network when $t$-th task arrives. (b) The controller implemented as a RNN to determine how many filters to add for the new task.

We use stochastic gradient descent to learn the newly added filters with $\eta$ as the learning rate,

$$W_t \backslash W_{t-1}^a \longleftarrow W_t \backslash W_{t-1}^a - \eta \nabla_{W_t \backslash W_{t-1}^a} L_t. \tag{5}$$

The expanded child network will be trained until the required number of epochs or convergence are reached. And then we test the child network on the validation dataset $\mathcal{V}_t$ and the corresponding accuracy $A_t$ will be returned. The parameters of the expanded network achieving the maximal reward (described in Section 3.3) will be the optimal ones for task $t$, and we store them for later tasks.

## 3.3 Reward Design

In order to facilitate our controller to generate better actions over time, we need design a reward function to reflect the performance of our actions. Considering both the validation accuracy and complexity of the expanded network, we design the reward for task $t$ by the combination of the two terms,

$$R_t = A_t(S_t, a_{1:m}) + \alpha C_t(S_t, a_{1:m}), \tag{6}$$

where $A_t$ represents the validation accuracy on $\mathcal{V}_t$, the network complexity as $C_t = -\sum_{i=1}^{m} k_i$, $k_i$ is the number of filters added in layer $i$, and $\alpha$ is a parameter to balance between the prediction performance and model complexity. Since $R_t$ is non-differentiable, we use policy gradient to update the controller, described in the following section.

## 3.4 Training Procedures

The controller's prediction can be viewed as a list of actions $a_{1:m}$, which means the number of filters added in $m$ layers , to design an new architecture for a child network and then be trained in a new task. At convergence, this child network will achieve an accuracy $A_t$ on a validation dataset and the model complexity $C_t$, finally we can obtain the reward $R_t$ as defined in Eq. (6). We can use this reward $R_t$ and reinforcement learning to train the controller.

To find the optimal incremental architecture the new task $t$, the controller aims to maximize its expected reward,

$$J(\theta_c) = V_{\theta_c}(s_t). \tag{7}$$

where $V_{\theta_c}$ is the true value function. In order to accelerate policy gradient training over $\theta_c$, we use actorcritic methods with a value network parameterized by $\theta_v$ to approximate the state value $V(s_t; \theta_v)$. The REINFORCE algorithm [16] can be used to learn $\theta_c$,

$$\nabla_{\theta_c} J(\theta_c) = E\left[\sum_{a_{1:m}} \pi(a_{1:m}|s_t, \theta_c)(R(s_t, a_{1:m}) - V(s_t, \theta_v))\frac{\nabla_{\theta_c}\pi(a_{1:m}|s_t, \theta_c)}{\pi(a_{1:m}|s_t, \theta_c)}\right]. \tag{8}$$

---
**Algorithm 1** RCL for Continual Learning
---
1: **Input:** A sequence of dataset $\mathcal{D} = \{\mathcal{D}_1, \mathcal{D}_2, \ldots, \mathcal{D}_T\}$
2: **Output:** $W_T^a$
3: **for** $t = 1, \ldots, T$ **do**
4:     **if** $t = 1$ **then**
5:         Train the base network using ( 3) on the first datasest $\mathcal{D}_1$ and obtain $W_1^a$.
6:     **else**
7:         Expand the network by Algorithm 2, and obtain the trained $W_t^a$.
8:     **end if**
9: **end for**
---

---
**Algorithm 2** Routine for Network Expansion
---
1: **Input:** Current dataset $\mathcal{D}_t$; previous parameter $W_{t-1}^a$; the size of action space for each layer $n_i, i = 1 \ldots, m$; number of epochs for training the controller and value network, $T_e$.
2: **Output:** Network parameter $W_t^a$
3: **for** $i = 1, \ldots, T_e$ **do**
4:     Generate actions $a_{1:m}$ by controller's policy;
5:     Generate $W_t^{(i)}$ by expanding parameters $W_{t-1}^a$ according to $a_{1:m}$;
6:     Train the expanded network using Eq. (5) to obtain $W_t^{(i)}$.
7:     Evaluate the gradients of the controller and value network by Eq. (9) and Eq.(10),

$$\theta_c = \theta_c + \eta_c \nabla_{\theta_c} J(\theta_c), \quad \theta_v = \theta_v - \eta_v \nabla_{\theta_v} L_v(\theta_v).$$

8: **end for**
9: Return the best network parameter configuration, $W_t^a = \text{argmax}_{W_t^{(i)}} R_t(W_t^{(i)})$.
---

A Monte Carlo approximation for the above quantity is,

$$\frac{1}{N} \sum_{i=1}^{N} \nabla_{\theta_c} \log \pi(a_{1:m}^{(i)} | s_t; \theta_c) \left( R(s_t, a_{1:m}^{(i)}) - V(s_t, \theta_v) \right). \tag{9}$$

where $N$ is the batch size. For the value network, we utilize gradient-based method to update $\theta_v$, the gradient of which can be evaluated as follows,

$$L_v = \frac{1}{N} \sum_{i=1}^{N} (V(s_t; \theta_v) - R(s_t, a_{1:m}^{(i)}))^2,$$

$$\nabla_{\theta_v} L_v = \frac{2}{N} \sum_{i=1}^{N} \left( V(s_t; \theta_v) - R(s_t, a_{1:m}^{(i)}) \right) \frac{\partial V(s_t; \theta_v)}{\partial \theta_v}. \tag{10}$$

Finally we summarize our RCL approach for continual learning in Algorithm 1, in which the sub-routine for network expansion is described in Algorithm 2.

### 3.5 Comparison with Other Approaches

As a new framework for network expansion to achieve continual learning, RCL distinguishes from progressive network [11] and DEN [17] from the following aspects.

- Compared with DEN, instead of performing selective retraining and network split, RCL keeps the learned parameters for previous tasks fixed and only updates the added parameters. Through this training strategy, RCL can totally prevent catastrophic forgetting due to the freezing parameters for corresponding tasks.

- Progressive neural networks expand the architecture with a fixed number of units or filters. To obtain a satisfying model accuracy when number of sequential tasks is large, the final complexity of progressive nets is required to be extremely high. This directly leads to high computational burden both in training and inference, even difficult for the storage of the

entire model. To handle this issue, both RCL and DEN dynamically adjust the networks to reach a more economic architecture.

- While DEN achieves the expandable network by sparse regularization, RCL adaptively expands the network by reinforcement learning. However, the performance of DEN is quite sensitive to the various hyperparameters, including regularization parameters and thresholding coefficients. RCL largely reduces the number of hyperparameters and boils down to only balancing the average validation accuracy and model complexity when the designed reward function. Through different experiments in Section 4, we demonstrate that RCL could achieve more stable results, and better model performance could be achieved simultaneously with even much less neurons than DEN.

## 4   Experiments

We perform a variety of experiments to access the performance of RCL in continual learning. We will report the accuracy, the model complexity and the training time consumption between our RCL and the state-of-the-art baselines. We implemented all the experiments in Tensorfolw framework on GPU Tesla K80.

**Datasets**   (1) MNIST Permutations [4]. Ten variants of the MNIST data, where each task is transformed by a fixed permutation of pixels. In the dataset, the samples from different task are not independent and identically distributed; (2) MNIST Mix. Five MNIST permutations $(P_1, \ldots, P_5)$ and five variants of the MNIST dataset $(R_1, \ldots, R_5)$ where each contains digits rotated by a fixed angle between 0 and 180 degrees. These tasks are arranged in the order $P_1, R_1, P_2, \ldots, P_5, R_5$. (3) Incremental CIFAR-100 [9]. Different from the original CIFAR-100, each task introduces a new set of classes. For the total number of tasks $T$, each new task contains digits from a subset of $100/T$ classes. In this dataset, the distribution of the input is similar for all tasks, but the distribution of the output is different.

For all of the above datasets, we set the number of tasks to be learned as $T = 10$. For the MNIST datasets, each task contains 60000 training examples and 10000 test examples from 10 different classes. For the CIFAR-100 datasets, each task contains 5000 train examples and 1000 examples from 10 different classes. The model observes the tasks one by one, and once the task had been observed, the task will not be observed later during the training.

**Baselines**   (1) **SN**, a single network trained across all tasks; (2) **EWC**, deep network trained with elastic weight consolidation [4] for regularization; (3) **GEM**, gradient episodic memory [7]; (4) **PGN**, progressive neural network proposed in [11]; (5) **DEN**, dynamically expandable network [17].

**Base network settings**   (1) Fully connected networks for MNIST Permutations and MNIST Mix datasets. We use a three-layer network with 784-312-128-10 neurons with RELU activations; (2) LeNet is used for Incremental CIFAR-100. LeNet has two convolutional layers and three fully-connected layers, the detailed structure of LeNet can be found in [5].

### 4.1   Results

We evaluate each compared approach by considering average test accuracy on all the tasks, model complexity and training time. Model complexity is measured via the number of model parameters after training all the tasks. We first report the test accuracy and model complexity of baselines and our proposed RCL for the three datasets in Figure 2.

**Comparison between fixed-size and expandable networks.**   From Figure 2, we can easily observe that the approaches with fixed-size network architectures, such as IN, EWC and GEM, own low model complexity, but their prediction accuracy is much worse than those methods with expandable networks, including PGN, DEN and RCL. This shows that dynamically expanding networks can indeed contribute to the model performance by a large margin.

**Comparison between PGN, DEN and RCL.**   Regarding to the expandable networks, RCL outperforms PGN and DEN on both test accuracy and model complexity. Particularly, RCL achieves

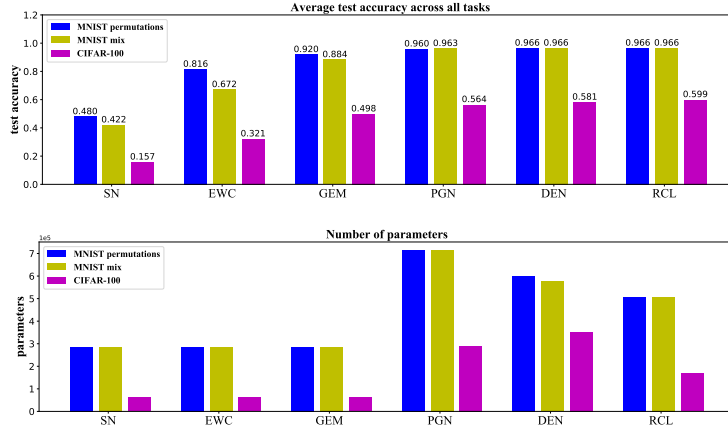

Figure 2: Top: Average test accuracy for all the datasets. Bottom: The number of parameters for different methods.

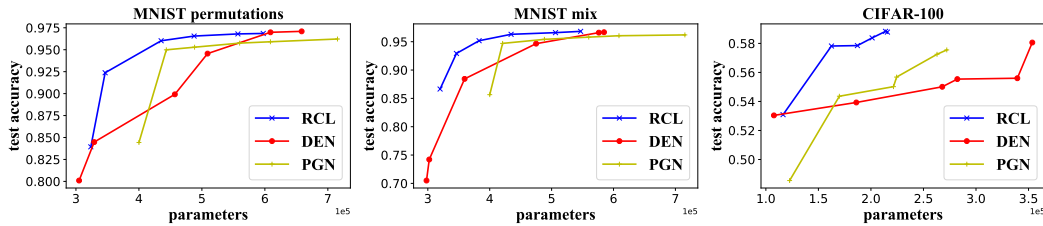

Figure 3: Average test accuracy *v.s.* model complexity for RCL, DEN and PGN.

significant reduction on the number of parameters compared with PGN and DEN, e.g. for incremental Cifar100 data, $42\%$ and $53\%$ parameter reduction, respectively.

To further see the difference of the three methods, we vary the hyperparameters settings and train the networks accordingly, and obtain how test accuracy changes with respect to the number of parameters, as shown in Figure 3. We can clearly observe that RCL can achieve significant model reduction with the same test accuracy as that of PGN and DEN, and accuracy improvement with same size of networks. This demonstrates the benefits of employing reinforcement learning to adaptively control the complexity of the entire model architecture.

**Comparison between RCL and Random Search.** We compare our policy gradient controller and random search controller on different datasets. In every experiment setup, hyper-parameters are the same except the controller (random search controller v.s. policy gradient controller). We run each experiment for four times. We found that random search achieves more than 0.1% less accuracy and almost the same number of parameters on these three datasets compared with policy gradient. We

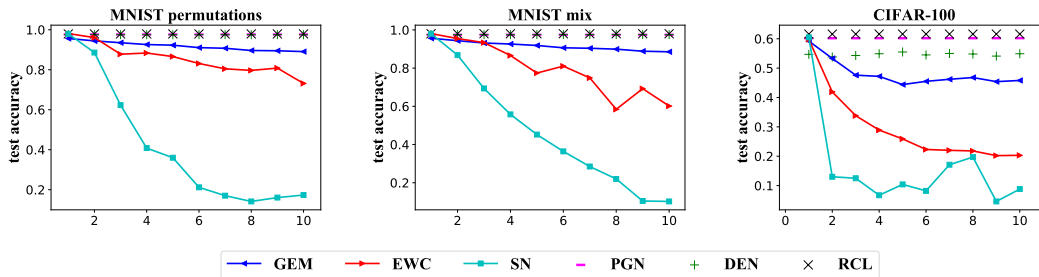

Figure 4: Test accuracy on the first task as more tasks are learned.

note that random search performs surprisingly well, which we attribute to the representation power of our reward design. This demonstrates that our well-constructed reward strikes a balance between accuracy and model complexity very effectively.

**Evaluating the forgetting behavior.** Figure 4 shows the evolution of the test accuracy on the first task as more tasks are learned. RCL and PGN exhibit no forgetting while the approaches without expanding the networks raise catastrophic forgetting. Moreover, DEN can not completely prevent forgetting since it retrains the previous parameters when learning new tasks.

**Training time** We report the wall clock training time for each compared method in Table 1). Since RCL is based on reinforcement learning, a large number of trials are typically required that leads to more training time than other methods. Improving the training efficiency of reinforcement learning is still an open problem, and we leave it as future work.

Table 1: Training time (in seconds) of experiments for all methods.

| Methods | IN | EWC | GEM | DEN | PGN | RCL |
|---|---|---|---|---|---|---|
| MNIST permutations | 173 | 1319 | 1628 | 21686 | 452 | 34583 |
| MNIST mix | 170 | 1342 | 1661 | 19690 | 451 | 23626 |
| CIFAR100 | 149 | 508 | 7550 | 1428 | 167 | 3936 |

**Balance between test accuracy and model complexity.** We control the tradeoff between the model performance and complexity through the coefficient $\alpha$ in the reward function (6). Figure 5 shows how varying $\alpha$ affects the test accuracy and number of model parameters. As expected, with increasing $\alpha$ the model complexity drops significantly while the model performance also deteriorate gradually. Interestingly, when $\alpha$ is small, accuracy drops much slower compared with the decreasing of the number of parameters. This observation could help to choose a suitable $\alpha$ such that a medium-sized network can still achieve a relatively good model performance.

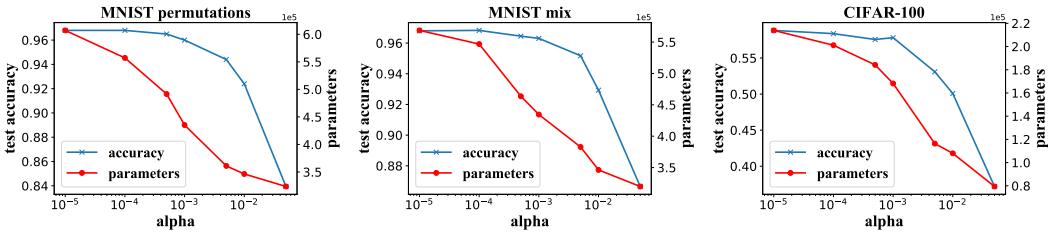

Figure 5: Experiments on the influence of the parameter $\alpha$ in the reward design

# 5 Conclusion

We propose a novel framework for continual learning, Reinforced Continual Learning. Our method searches for the best neural architecture for coming task by reinforcement learning, which increases its capacity when necessary and effectively prevents semantic drift. We implement both fully connected and convolutional neural networks as our task networks, and validate them on different datasets. The experiments demonstrate that our proposal outperforms the exiting baselines significantly both on prediction accuracy and model complexity.

As for future works, two directions are worthy of consideration. Firstly, we will develop new strategies for RCL to facilitate backward transfer, i.e. improve previous tasks' performance by learning new tasks. Moreover, how to reduce the training time of RCL is particularly important for large networks with more layers.

## Acknowledgments

Supported by National Natural Science Foundation of China (Grant No: 61806009) and Beijing Natural Science Foundation (Grant No: 4184090).

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
