[Supplementary Material]

# Supplementary material for:
# Reinforced Continual Learning

**Ju Xu**
Center for Data Science, Peking University
Beijing, China
xuju@pku.edu.cn

**Zhanxing Zhu** [*]
Center for Data Science, Peking University &
Beijing Institute of Big Data Research (BIBDR)
Beijing, China
zhanxing.zhu@pku.edu.cn

## 1   Experiment settings

In this section, we will present the experiments details of our model and baselines. When dealing with dataset MNIST permutations and dataset MNIST mix, we use a three-layer network with 784-312-128-10 neurons, and the learning rate is 0.001, the batch size is 32, the training epochs are 15 for all models. When expanding the network, the size of search space is 30 across all layers for RCL,DEN and PGN. As for CIFAR-100, we use LeNet as our task network. The training epochs are 20 and the learning rate is 0.001. The search space is 5 in convolutional layers, 25 in fully-connected layers for RCL,DEN and PGN.

Our controller is implemented as a LSTM network. The LSTM network has two layers, and the hidden size is 100. Our value network is implemented as a fully-connected network, which has only one layer. The learning rate for our controller is 0.001, for our value network is 0.005.

The $\alpha$ in our reward design is 0.0003 for MNIST permutations, 0.0002 for MNIST mix, and 0.001 for dataset CIFAR-100. The l1_lambda is 0.00001, l2_lambda is 0.0001, gl_lambda is 0.001, regular_lambda is 0.5, loss_thr is 0.01, spl_thr is 0.05 in DEN for MNIST permutations and MNIST mix. As for CIFAR-100, the hyperparameters in DEN is the same except regular_lambda is 5.

---
[*]Corresponding author.