[Reviews · NeurIPS 2018]

Reviewer 1



Summary: This paper proposes a way to deal with "catastrophic forgetting" by assigning the parts of the model specifically for particular tasks that are being learned and learn to grow the network by adding more filters as more and more task are introduced. In this way parts of the model become more task-specific. The controller which decides how many filters or units to add to each layer is an LSTM network over the layers that learns the sequential relation between consecutive layers. They train the controller via reinforcement learning using REINFORCE with learned actor and critic. They have done a smart reward shaping in order to train the controller. Overall Comments: The idea of adaptively expanding the network according to the new task that is being added and the capacity that is assigned by the controller is task dependent is very neat. However, I am quite surprised that the REINFORCE objective actually worked in this setup. Because in the similar settings for instance in adaptive computation time, REINFORCE has not really worked that well. I think the main reason that it worked in this setup might be the reward shaping. I think overall the paper is well-written besides some small typos. For example "s/matche/matches/" in line 24. I think one main part that this paper lacks is the experiments section: 1) Experiments are only limited to CIFAR10 and MNIST. These are very toyish tasks. I would be in particular interested in seeing results on some NLP or reinforcement learning task. For example on one of the tasks that Progressive neural networks were tested on. 2) More ablations are needed. In particular, ablations on the controller would be super useful. For example, one ablation that I would like to see would be a fixed controller or a random controller would be interesting as well. Overall, I like the paper and the idea, but I don't buy the experiments part of the paper completely.

Reviewer 2



The work gave a nice application of RL to the continual learning problem, particularly focusing on the forward transfer learning case. Namely, as in Progressive Net or DEN, the proposed method follows to expand model architecture for the new task. But, unlike Progressive Net, which expands with fixed size network, and DEN, which has various hyperparameters to tune, the proposed method applies RL framework to learn the model expanding step for each task. The specific RL technique is not necessarily novel, but quite standard - LSTM controller and actor-critic method for learning. However, I think the idea of applying RL to continual learning is novel enough for a publication. In their experimental results, RCL achieves essentially the same accuracy as PGN and DEN, but with fewer parameters and hyperparameters to tune. The downside is that the training time is much longer than the other method. Also, since their model complexity indeed grows with the number of tasks, it cannot handle too many tasks, which is the common limitation of PGN and DEN. The result on the forgetting behavior is not too surprising since they are freezing the network for the older tasks. Hence, there is not backward transfer happening, if the data from the old task arrives again.

Reviewer 3



This paper presents a simple but effective way to dynamically grow a neural network in a continual learning setup. The framework uses reinforcement learning to learn a policy that outputs decisions in terms of number of nodes/filters to add, and those additional filters are trained specifically for the new task (while the previous layers are held constant to prevent reduction in performance for old tasks). This is progressively done layer by layer where previous decisions for earlier layers serve as input to an LSTM in order to make decisions for the subsequent layer. Results show similar performance to approaches such as DEM, but have some better results for CIFAR-100 especially and the method results in networks with lower complexity. Strengths - Good overview of two groups of approaches on this topic - The idea of using RL for continual learning is a good one, and one that makes sense given recent advancements in RL for architecture search. - The results seem positive, especially for CIFAR-100 and for lowering the overall complexity of the solutions. - The method is simple, which means that it should be easy to implement, barring issues in RL training stability. Releasing source code would be very helpful in this regard for reproducing the results in the paper. Weaknesses - An argument against DEN, a competitor, is hyper-parameter sensitivity. First, this isn't really shown, but second (and more importantly) reinforcement learning is well-known to be extremely unstable and require a great deal of tuning. For example, even random seed changes are known to change the behavior of the same algorithm, and different implementation of the same algorithm can get very different results (this has been heavily discussed in the community; see keynote ICLR talk by Joelle Pineau as an example). This is not to say the proposed method doesn't have an advantage, but the argument that other methods require more tuning is not shown or consistent with known characteristics of RL. * Related to this, I am not sure I understand experiments for Figure 3. The authors say they vary the hyper-parameters but then show results with respect to # of parameters. Is that # of parameters of the final models at each timestep? Isn't that just varying one hyperparameter? I am not sure how this shows that RCL is more stable. - Newer approaches such as FearNet [1] should be compared to, as they demonstrated significant improvement in performance (although they did not compare to all of the methods compared to here). [1] FearNet: Brain-Inspired Model for Incremental Learning, Ronald Kemker, Christopher Kanan, ICLR 2018. - There is a deeper tie to meta-learning, which has several approaches as well. While these works don't target continual learning directly, they should be cited and the authors should try to distinguish those approaches. The work on RL for architecture search and/or as optimizers for learning (which are already cited) should be more heavily linked to this work, as it seems to directly follow as an application to continual learning. - It seems to me that continuously adding capacity while not fine-tuning the underlying features (which training of task 1 will determine) is extremely limiting. If the task is too different and the underlying feature space in the early layers are not appropriate to new tasks, then the method will never be able to overcome the performance gap. Perhaps the authors can comment on this. - Please review the language in the paper and fix typos/grammatical issues; a few examples: * [1] "have limitation to solve" => "are limited in their ability to solve" * [18] "In deep learning community" => "In THE deep learning community" * [24] "incrementally matche" => "incrementally MATCH" * [118] "we have already known" => "we already know" * and so on Some more specific comments/questions: - This sentence is confusing [93-95] "After we have trained the model for task t, we memorize each newly added filter by the shape of every layer to prevent the caused semantic drift." I believe I understood it after re-reading it and the subsequent sentences but it is not immediately obvious what is meant. - [218] Please use more objective terms than remarkable: "and remarkable accuracy improvement with same size of networks". Looking at the axes, which are rather squished, the improvement is definitely there but it would be difficult to characterize it as remarkable. - The symbols in the graphs across the conditions/algorithms is sometimes hard to distinguish (e.g. + vs *). Please make the graphs more readable in that regard. Overall, the idea of using reinforcement learning for continual learning is an interesting one, and one that makes sense considering recent advances in architecture search using RL. However, this paper could be strengthened by 1) Strengthening the analysis in terms of the claims made, especially with respect to not requiring as much hyper-parameter tuning, which requires more evidence given that RL often does require significant tuning, and 2) comparison to more recent methods and demonstration of more challenging continual learning setups where tasks can differ more widely. It would be good to have more in-depth analysis of the trade-offs between three approaches (regularization of large-capacity networks, growing networks, and meta-learning). ============================================== Update after rebuttal: Thank you for the rebuttal. However, there wasn't much new information in the rebuttal to change the overall conclusions. In terms of hyper-parameters, there are actually more hyper-parameters for reinforcement learning that you are not mentioning (gamma, learning rate, etc.) which your algorithm might still be sensitive to. You cannot consider only the hyper-parameter related to the continual learning part. Given this and the other limitations mentioned, overall this paper is marginally above acceptance so the score has been kept the same.